# Assessment of frailty in patients with advanced chronic kidney disease, and the role of microvesicles: A single-center study

Rocio Gimena Muñoz[1]*, Gemma Valera Arévalo[2],
María del Mar Rodríguez San Pedro[2], María Pérez Fernández[1],
Juan Arévalo Serrano[3], Sushrut S. Waikar[4], Rafael Ramírez-Chamond[5,6],
Diego Rodríguez Puyol[1,5,7], Julia Carracedo[2☙], Patricia Martinez Miguel[1☙]

**1** Service of Nephrology, University Hospital Príncipe de Asturias, Alcalá de Henares, Spain, **2** Genetics and Physiology Department, Science Faculty, Complutense University, Investigation Institute Hospital 12 de Octubre (imas12), Madrid, Spain, **3** Service of Internal Medicine, University Hospital Príncipe de Asturias, Faculty of Medicine and Health Sciences, University of Alcalá, Alcalá de Henares, Spain, **4** Boston University School of Medicine, Nephrology Department, Boston Medical Center, Boston, Massachusetts, United States of America, **5** Ramón & Cajal Investigation Institute (IRYCIS), Madrid, Spain, **6** Systems Biology Department, University of Alcalá, Alcalá de Henares, Madrid, Spain, **7** Medicine Department, University of Alcalá, Alcalá de Henares, Madrid, Spain

☙ Both authors have contributed equally to the direction of this work.
* rociogimu3@gmail.com

## Abstract

### Background

Frailty is a highly prevalent syndrome in patients with advanced age and chronic diseases, and it is associated with atherothrombotic pathologies, suggesting a procoagulant state in these patients. Circulating microvesicles (cMVs), are small phospholipid-rich vesicles, which have been shown to participate in atherothrombotic onset and progression. We aim to analyze frailty in patients with advanced chronic kidney disease (CKD), and to determine the role of microvesicles in this population.

### Materials and methods

We conducted a prospective cohort study with 85 patients with advanced CKD. Our main objective was to evaluate frailty and its novel association with circulating microvesicles in advanced CKD. To define frailty, Fried's five criteria were used and we obtained blood for cMVs analysis by flow cytometry.

### Results

The prevalence of frailty in patients with advanced CKD was 27% (CI 95% 17-37%). We found that risk factors for frailty were age (OR 1.06; CI 95% to 1.11; p=0.027), type 2 diabetes mellitus (OR 5.77; CI 95% to 18.9; p=0.004) and hemoglobin (g/dL) (OR 0.63; CI 95% to 0.94; p=0.023). Total cMVs, platelet derived cMVs and

**Data availability statement:** All relevant data are within the manuscript and its Supporting information files.

**Funding:** The author(s) received no specific funding for this work.

**Competing interests:** The authors have declared that no competing interests exist.

endothelial derived cMVs were significantly higher in frail patients. In the predictive multivariable binary logistic regression model over frailty, three predictors explain 41% of the variability of frailty (Nagelkerke square R = 0.41; p <0.001) with the following contribution: T2DM (19%), total cMVs (14%) and hemoglobin (8%).

## Discussion

Frailty is highly prevalent in patients with advanced CKD. Although we still do not know in depth the mechanisms involved in frailty, to our knowledge this is the first study that links cMVs and frailty in patients with advanced CKD. In our opinion it could be explored as a good biomarker or therapeutic target in advanced CKD.

## Introduction

Frailty affects approximately 10% of community-dwelling older adults, increasing with age [1]. It associates high risk for adverse health outcomes, including mortality, institutionalization, falls, and hospitalization [2–5]. Geriatricians define frailty as a biologic syndrome of decreased reserve and resistance to stressors, resulting in increased risk of morbidity and mortality [6–8]. However, frailty is not only seen in old age, but rather it is associated with many chronic diseases, such as chronic kidney disease (CKD). In fact, frailty is identifiable in approximately 35% of patients with CKD G3-5 (eGF < 60mL/min/1.73m2) [9], a percentage similar to that found in patients with advanced CKD (eGFR < 20mL/min/1.73m2) [10] and patients undergoing hemodialysis [11,12]. Frailty is associated with higher risks of hospitalization and mortality [13–15]. Evidence also shows that advanced age and the presence of diabetes mellitus increases the risk of frailty in patients undergoing hemodialysis [16]. Many risk factors are likely related to frailty, but there is still a need to understand these in a more thorough manner. In contrast with the knowledge about frailty in dialysis patients, it is still unclear what the prevalence is in patients with end stage renal disease not undergoing renal replacement therapy, suggesting the need for a more exhaustive study in this population.

Frail older adults present a high prevalence of cardiovascular diseases (CVD) and atherothrombotic pathologies [16], suggesting a procoagulant state in many of these patients. Circulating microvesicles (cMVs), are small phospholipid-rich vesicles involved in intercellular signaling pathways, which have been shown to participate in atherothrombotic onset and progression [17]. CD142, also known as platelet tissue factor, is an essential protein involved in the clotting process. Together with cMVs, CD142 will take part in most thrombotic events, as they enhance thrombin formation [18–20]. As shown by Arauna and colleagues, frail older adults presented higher concentrations of platelet derived cMVs, compared to robust age-matched older adults [21], suggesting a prothrombotic phenotype in these patients. Endothelial derived cMVs also contribute to atherosclerosis development by promoting endothelial dysfunction and thrombus formation after rupture [17]. There is strong evidence of uremic patients having a high thrombotic tendency, and high incidence of thrombotic

events [22,23]. We aim to analyze frailty in patients with advanced CKD, and to determine the role of microvesicles in this population.

## Materials and methods

We conducted a single-center, observational, prospective cohort study involving 85 patients with advanced CKD from our hospital. Patients were included from December 2019 till March 2022. The study was conducted according to the Declaration of Helsinki. We obtained authorization and approval of "The Research Ethics Committee of the Príncipe de Asturias University Hospital". All of the patients were carefully informed about the study and provided written informed consent (which they signed in order to be included). Inclusion criteria included: adults>18 years old, eGFR<20 ml/min. Exclusion criteria included: patients undergoing renal replacement therapy or patients who decide not to initiate renal replacement therapy and were receiving only medical management without dialysis.

Freid's five criteria (asthenia, weight loss, loss of strength, slow gait, and decreased physical activity) were used to define frailty [1]. Frailty phenotype was identified by the presence of three or more of Freid's five criteria; grip strength (according to gender and BMI), gait speed<0.8 m/s on 4 m test, 3 or more days of asthenia per week (assessed via patient self-report using a structured symptom questionnaire), reported physical activity<2.5h per week and reported unintentional weight loss >10 pounds over the past year.

We obtained complete lab tests (basic routine panel, renal function panel and complete blood count) at the time of testing. Blood was also drawn in tubes with ethylenediaminetetraacetic (EDTA) and these were sent to the biobank for the microvesicle analysis. Clinical data was extracted from the clinical history. There was a median follow-up time of 18 months, and during this time the start of dialysis (both haemodialysis and peritoneal dialysis) was evaluated, as well as mortality.

The total number of cMVs (AnnexinV+) platelet-derived MVs (AnnexinV+CD31+CD41+) and endothelial-derived MVs (AnnexinV+CD31+CD41–), as well as the expression of tissue factor (CD142) were determined as previously described [24–26]. A quadruple-staning immunofluorescence technique was used. Monoclonal antibodies conjugated with fluorochromes against Annexin V (Annexin V-FITC Kit; MiltenyiBiotec, BergischGladbach, Germany), CD41/integrin subunit alpha 2b (MEM-06 clone, peridnine chlorophyll protein; Invitrogen), CD31/platelet and endothelial cell adhesion molecule 1 [PECAM1] (WM-59 clone, phycoerythrin; BD Bioscience), and CD142/tissue factor (HTF-1 clone, allophycocyanin [APC]; Invitrogen) were used. Platelet-free plasma samples were centrifuged at 11,000×g for 20min and resuspended in Annexin-V binding buffer (Annexin V-FITC Kit; MiltenyiBiotec). Subsequently, the samples were incubated with the corresponding antibodies for 40min at room temperature in darkness, fixed using Cell Fix (BD Bioscience), and stored at 4°C until assessment, within the following 24h. The cMVs subpopulations were characterized through flow cytometry using a FACSAria™ III cytometer (BD Biosciences) with the support of the staff of the cytometry associated research center of Complutense University of Madrid (Spain) and analyzed by the FlowJoTM software. The standardization on the FACSAria™ III device was carried out as previously described [26].

Student's t test was used for continuous variables with normal distribution, and Mann-Whitney U test for continuous variables with non-parametric distribution. For categorical variables chi-square or Fisher's exact test were used. To define the risk factors associated with frailty, we conducted a univariate and multivariate analysis using estimative binary logistic regression models. The median follow up was calculated by the inverse Kaplan-Meier. The start of dialysis (both haemodialysis and peritoneal dialysis) was evaluated, using multivariable (controlled by T2DM, hemoglobin and phosphorus) Cox proportional hazards models.Variables for the multivariable Cox models were selected based on clinical relevance and statistical trends in univariate analysis. We use univariate Cox proportional hazard models for the calculation of the hazards ratios (HR) and P values of mortality (only 4 deaths did not permit multivariate adjustment). Pearson's correlation coefficient was used to evaluate correlations between cMVs and different parameters. Statistical analysis was performed using IBM SPSS Statistics 26.00 (IBM Corp, Armonk, NY, USA). P<0.05 was considered statistically significant.

## Results

The prevalence of frailty in patients with advanced CKD was 27% (CI 95% 17–37%). The age range of the sample was from 38 to 86 years. Frail patients were older, with more type 2 diabetes mellitus (T2DM) and higher cardiovascular disease as shown in Table 1. Laboratory findings of frail and non-frail patients are also shown in Table 1, highlighting the fact that frail patients had a lower eGFR and hemoglobin level, and a higher phosphorus level, with no differences in PTH and calcium levels. Frail patients had higher concentrations of total cMVs.

To try to assess risk factors related to frailty, we conducted a univariate and multivariate analysis using estimative binary logistic regression models. Each of the variables of interest were adjusted by two predictor variables (limited by the number of cases) as shown in Table 2. A predictive or explicative multivariable binary logistic regression model over frailty was made with the predictors age, T2DM, hemoglobin and total cMVs. We found out that the adjusted independent risk factors for frailty were age, T2DM and anemia (low hemoglobin).

Each variable of interest is adjusted by two predictor variable (only 2 predictor variables are used, limited to the number of cases, as potential confusion factors): Age (T2DM and hemoglobin), T2DM (age and hemoglobin), CVD (age and hemoglobin), hemoglobin (treatment with erythropoiesis-stimulating agents and treatment with iron), phosphorus (age and treatment with phosphate binders), eGFR-estimated glomerular filtration rate (age and T2DM), ferritin (treatment with erythropoiesis-stimulating agents and treatment with iron), bicarbonate (age and T2DM), PTH-parathyroid hormone (age and treatment with phosphate binders).

**Table 1. Demographic, clinical features and laboratory findings of frail and non-frail patients with advanced chronic kidney disease.**

|  | Non-Frail | Frail | P value |
|---|---|---|---|
| N (% of individuals) | 62 (73) | 23 (27) |  |
| Female n(%) | 20 (67) | 10 (33) | 0.336 |
| Male n(%) | 42 (76) | 13 (24) |  |
| Age, years (mean±SD) | **66±11.9** | **73±10.6** | **0.022** |
| T2DM n(%) | **28 (45%)** | **19 (83%)** | **0.002** |
| CVD n(%) | **17 (27%)** | **12 (52%)** | **0.032** |
| HBP n(%) | 58 (94%) | 21(91%) | 0.660 |
| Creatinine (mg/dL) | 4.3±1.6 | 4.9±2.0 | 0.168 |
| eGFR (mL/min/1.73m2) | **13.5±4.8** | **11.4±5.5** | **0.045** |
| Hemoglobin (g/dL) | **12.0±1.4** | **11.1±1.4** | **0.018** |
| Platelets (mcL) | $205 \cdot 10^3 \pm 62 \cdot 10^3$ | $226 \cdot 10^3 \pm 89 \cdot 10^3$ | 0.224 |
| Transferrin Saturation Index (%) | 26.7±8.7 | 27.9±17.1 | 0.462 |
| Ferritin (ng/mL) | 205±164 | 305±329 | 0.435 |
| Blood pH | 7.28±0.06 | 7.28±0.06 | 0.934 |
| Bicarbonate (HCO3-) (mmol/L) | 20.8±3.4 | 20.2±3.0 | 0.520 |
| Calcium (serum) (mg/dL) | 9.1±0.8 | 8.8±0.7 | 0.101 |
| Phosphorus (mg/dL) | **4.6±1.2** | **5.1±0.7** | **0.004** |
| PTH (pg/mL) | 336±217 | 343±163 | 0.535 |
| LDL (mg/dL) | 148±35 | 139±31 | 0.400 |
| CRP (mg/dL) | 5.2±7.5 | 5.9±6.4 | 0.499 |
| Interleukin 6 (IL-6) | 11±14.5 | 11±6.4 | 0.363 |
| Total microvesiclesx$10^{-4}$ | 14.36±12.51 | 23.75±12.66 | 0.002 |

T2DM (type 2 diabetes mellitus), CVD (cardiovascular disease), HBP (high blood pressure),eGFR (estimated glomerular filtration rate), PTH (parathyroid hormone), LDL (low-density lipoprotein), CRP (c-reactive protein).

**Table 2. Risk Factors related to Frailty in patients with advanced chronic kidney disease.**

| | Univariate | | Multivariate* | |
|---|---|---|---|---|
| | OR (95%CI) | p value | OR(95% CI) | p value |
| Age (years) | 1.06 (1.01, 1.11) | **0.027** | 1.06 (1.01, 1.11) | **0.027** |
| T2DM (type 2 diabetes mellitus) | 5.77 (1.76, 18.9) | **0.004** | 5.77 (1.76, 18.9) | **0.004** |
| CVD (cardiovascular disease) | 2.89 (1.07, 7.77) | **0.036** | 2.40 (0.86, 6.67) | 0.094 |
| Hemoglobin (g/dL) | 0.63 (0.42, 0.94) | **0.023** | 0.63 (0.42, 0.94) | **0.023** |
| Phosphorus (mg/dL) | 1.54 (0.98, 2.42) | 0.060 | 1.54 (0.98, 2.42) | 0.060 |
| eGFR (mL/min/1.73m2) | 0.91 (0.82, 1.01) | 0.083 | 0.91 (0.82, 1.01) | 0.083 |
| Ferritin (ng/mL) | 1.002 (1.000, 1.004) | 0.122 | 1.002 (1.000, 1.004) | 0.122 |
| Bicarbonate(mmol/L) | 0.95 (0.82, 1.11) | >0.200 | 0.95 (0.82, 1.11) | >0.200 |
| PTH(pg/mL) | 1.00 (0.99, 4.01) | >0.200 | 1.000 (0.997, 1.003) | >0.200 |

After a median follow up of 18 months (CI 95% 14–22), the start of dialysis (both haemodialysis and peritoneal dialysis) was evaluated, as well as mortality, using multivariable Cox proportional hazard models. 23 patients (27.1%) began dialysis. Fig 1a shows survival curves by Kaplan-Meier over the start of dialysis (univariate log-rank test 0,126). Estimative multivariate Cox proportional hazard models controlled by T2DM, hemoglobin and phosphorus, showed that frailty did not influence the start of dialysis (HR 0.77; 95% CI 0.32 to 1.85; p=0.557). On the other hand, using univariate Cox proportional hazard models (only 4 deaths [4.1%] did not permit multivariate adjustment), showed that frailty was related with mortality as shown in (Fig 1b) (HR 3.61; 95% CI 1.03 to 12.72; p=0.045).

Total cMVs (p=0.002), platelet derived cMVs (p=0.001) and endothelial derived cMVs (p=0.024) were also evaluated in frail and non-frail patients. All of these were significantly higher in frail patients, (Fig 2). We also measured CD142 (tissue factor) as a procoagulant marker, in both platelet (p=0.004) and endothelial (p=0.014) derived cMVs. It was significantly higher in frail patients, compared to non-frail patients (Fig 3).

There was a correlation between serum platelets and total cMVs ($r_s$=0.30; p=0.007), as well as, platelet derived cMVs ($r_s$=0.32; p=0.004) (shown in Table 3 and Fig 4).

In the explicative or predictive multivariable binary logistic regression model over frailty, three predictors (T2DM, hemoglobin and total cMVs) explain 41% of the variability of frailty (Nagelkerke square R=0.41; p<0.001) with the following contribution of each variable: T2DM (19%), total cMVs (14%) and hemoglobin (8%) (Table 4).

## Discussion

Frailty is a prevalent syndrome, specially related to old age but also present in chronic kidney disease. The prevalence of frailty in patients with CKD is around 35% [9]. In our cohort, the prevalence of frailty in patients with advanced CKD was 27% (CI 95% 17–37%), more than in the general geriatric population (around 10%) [2,27,28]. We were also able to observe that frail patients had a slightly decreased eGFR, with significantly higher phosphorus levels, with no differences in parathyroid hormone and calcium levels (Table 1). The higher phosphorus levels could be related to the decrease in eGFR, but also due to altered bone-mineral metabolism. Frail patients seem to have lower Klotho values [28], and this could be related to phosphorus metabolism. Nevertheless, in the multivariate analysis, phosphorous did not behave as a risk factor for frailty. Other studies have shown that risk factors related to frailty in uremic patients include advanced age, T2DM and anemia [15,16]. This was also true in our advanced CKD cohort. The comparison of frailty by age is shown in Table 1 and was statistically significant in both the univariate and multivariate analyses (OR 1.06; 95% CI 1.01 to 1.11; p=0.027). In our sample, each additional year of age increased the risk of frailty by an average of 6%.Frailty increases morbidity and mortality and is closely related to cardiovascular diseases and atherothrombotic pathologies [16]. We found that frail patients with advanced CKD have a higher mortality, as others have also found [29]. Therefore, frailty could be

a)

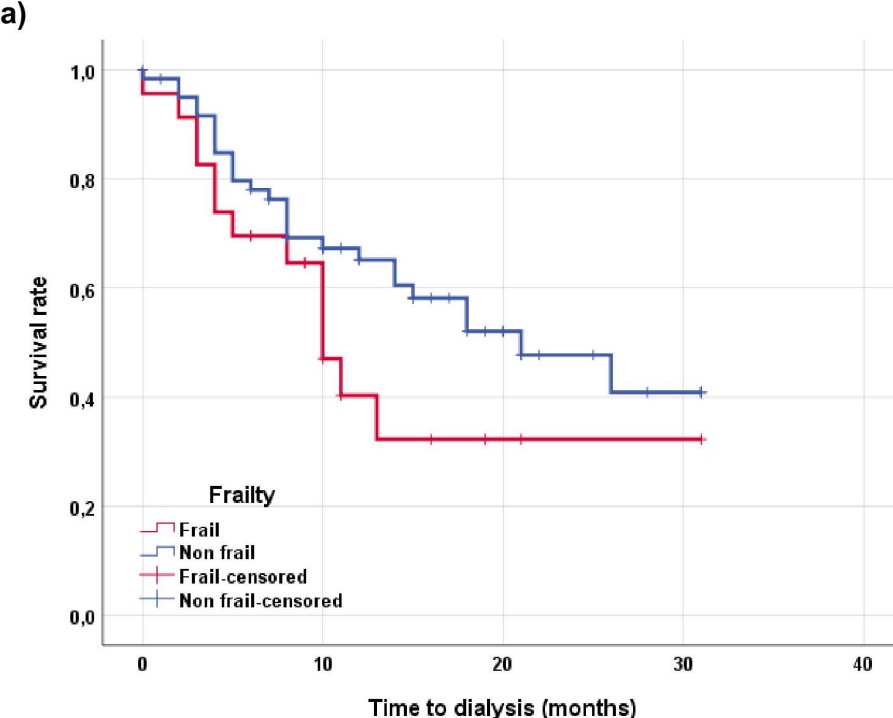

b)

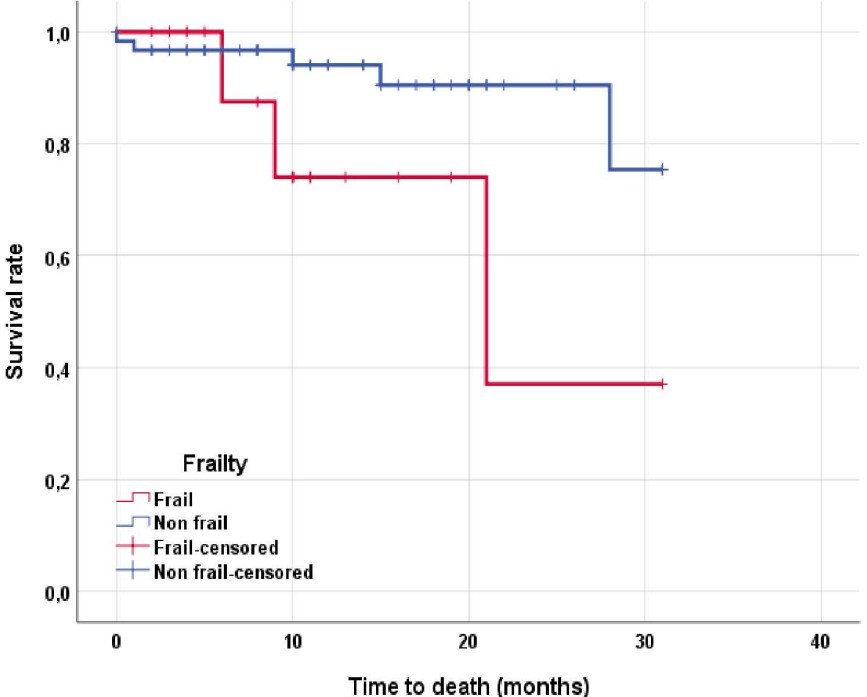

**Fig 1. Survival curves.** Kaplan-Meier survival curves of frail and non-frail patients with advanced chronic kidney disease(a)Time to dialysis in frail compared to non-frail patients. (b)Time to death in frail compared to non-frail patients.

**A) Total cMVs**

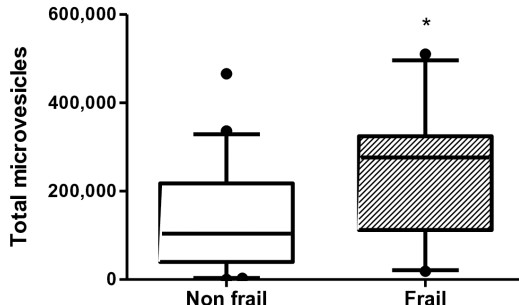

**B) Platelet cMVs**

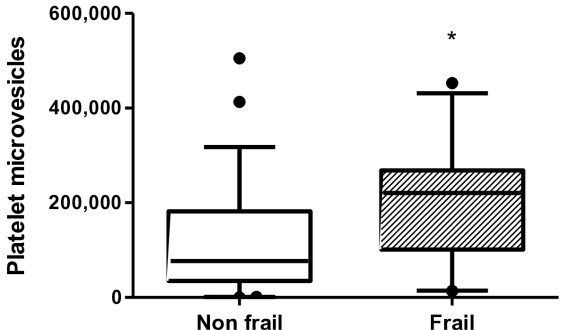

**C) Endothelial cMVs**

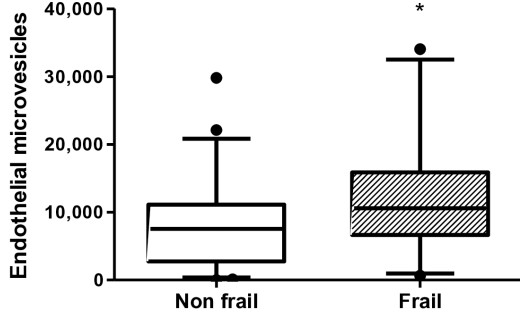

**Fig 2. Circulating microvesicles (cMVs) concentration (mL) in frail and non-frail patients with advanced chronic kidney disease.** The box and whisker graphs show the median and percentile 10–90. Concentration of total cMVs according to frailty status p = 0.002 (a) Concentration of platelet cMVs according to frailty status p = 0.001 (b) Concentration of endothelial cMVs according to frailty status p = 0.024 (c).

a good index for evaluation of vulnerable patients and early screening a useful tool. In fact, there is an ongoing need to identify other useful biomarkers associated with unhealthy aging, frailty and those at high risk of cardiovascular disease [30,31].

cMVs are small cell-derived, phospholipid-rich vesicles, involved in intercellular signaling pathways. cMVs have been implicated in physiological processes, but also in pathophysiological processes. They are involved in the development of inflammatory effects, thrombosis, cell proliferation and calcification in the vasculature observed in pathological conditions [32–34]. cMVs are carriers of damage-associated mediators, cytokines and enzymes that play a critical role in

## A) Platelet microvesicles CD142+

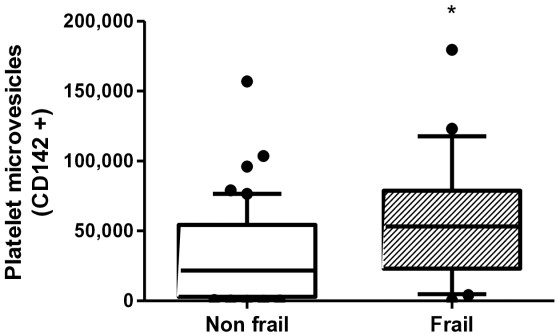

## B) Endothelial microvesicles CD142+

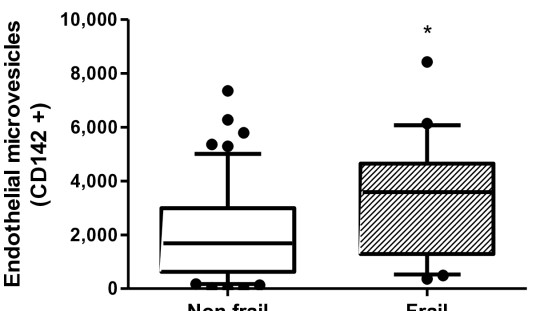

**Fig 3. Circulating microvesicles with CD142 (mL) in Frail patients compared to Non-Frail patients.** Platelet microvesicles CD142+ in frail and non-frail patients (p = 0.004) (a). Endothelial microvesicles CD142+ in frail and non-frail patients (p = 0.014) (b). The box and whisker graphs show the median and percentile 10–90.

**Table 3. Correlation between circulating microvesicles in patients with advanced chronic kidney disease and different parameters.**

|  | Total microvesicles | Endothelial microvesicles | Platelet microvesicles |
|---|---|---|---|
| Age (years) | $r_s$: 0.20<br>p: 0.07 | $r_s$: 0.26<br>p: 0.02 | $r_s$: 0.19<br>p: 0.08 |
| LDL (low-density lipoprotein) | $r_s$: 0.046<br>p: 0.69 | $r_s$: −0.09<br>p: 0.43 | $r_s$: 0.051<br>p: 0.66 |
| Platelets (mcL) | $r_s$: 0.30<br>p: 0.007 | $r_s$: 0.12<br>p: 0.29 | $r_s$: 0.32<br>p: 0.004 |

$r_s$: Spearman's Rho correlation coefficient.

p: p of significance.

inflammatory diseases [35]. The elevation of plasma cMVs, specially of endothelial origin, is considered a marker of vascular dysfunction [36]. On the other hand, high concentrations of platelet derived cMVs, enhance platelet deposition and thrombus formation [37], and they are also a link between atherosclerosis and thrombosis [38]. Elevated levels of cMVs and CD142 carrying cMVs have been observed in patients with acute myocardial infarction and unstable angina and are related to the severity of these [39]. Frail older adults have a higher concentration of platelet derived cMVs, compared to robust age-matched older adults [21].

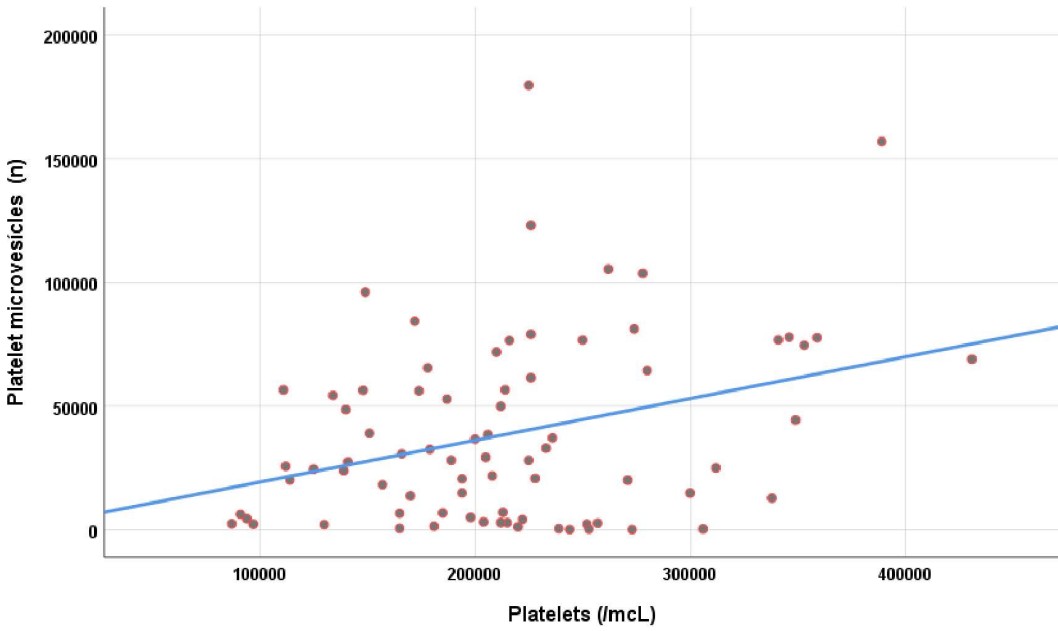

**Fig 4. Correlation between platelets and platelet microvesicles.**

**Table 4. Multivariate predictive model of Frailty with total microvesicles in patients with advanced chronic kidney disease.**

|  | OR | 95% CI | P Value | R2 Nagelkerke |
|---|---|---|---|---|
| Global model |  |  | <0.001 | 0.41 |
| Total micovesicles x $10^{-4}$ | 1.08 | 1.02–1.13 | 0.004 | 0.14 |
| T2DM | 7.79 | 2.00–30.24 | 0.003 | 0.19 |
| Hemoglobin (g/dL) | 0.60 | 0.38–0.94 | 0.027 | 0.08 |

T2DM (type 2 diabetes mellitus).

While frailty in CKD patients is well documented, our study provides novel insight by demonstrating a significant association between increased levels of circulating microvesicles and frailty. Frail patients with advanced CKD had higher concentrations of both endothelial- and platelet-derived cMVs, and these cMVs also expressed more tissue factor (CD142), supporting a potential procoagulant phenotype. We could also observe the correlation between the number of platelets and the amount of total and platelet derived cMVs, which supports the involvement of these elements in thrombus formation (Table 3 and Fig 4).These findings suggest that cMVs are not only markers of endothelial and platelet activation but may also reflect underlying pathophysiological mechanisms linking frailty with cardiovascular risk in CKD. Importantly, in our multivariable predictive model, total cMVs explained a meaningful proportion of frailty variability, independent of traditional factors such as age and T2DM. To our knowledge, this is the first study to establish such a link in this population. As such, cMVs may serve as promising biomarkers or therapeutic targets for frailty prevention and management in advanced CKD.

Although there are studies that report the association of cMVs with classic cardiovascular risk factors, including dyslipidemia [40], in our study we did not find a correlation between cMVs and the amount of low-density lipoprotein (LDL). It should be noted that frail patients (with unintentional weight loss as a criteria), and patients with CKD have a different lipid profile, and that it is not so much the amount of LDL as the proinflammatory capacity of LDL that may contribute to atherothrombotic damage [41].

Limitations of this study include its small sample size, which means that results such as the prevalence of frailty in the subpopulation of patients with advanced CKD cannot be generalized. Also, the analysis using estimative binary logistic regression models can be adjusted for a limited number of predictor variables. Even so, our results are concordant with other published studies, and are novel in some aspects not previously evaluated, such as the increase in cMVs in frail patients with advanced CKD.

According to this study, we can conclude that frailty could be highly prevalent in patients with advanced CKD, with advanced age, T2DM and anemia being possible risk factors for the development of this syndrome. We can also state, that although we still do not know in depth the mechanisms involved in frailty, to our knowledge this is the first study that links cMVs and frailty in patients with advanced CKD, suggesting this could be explored as a good biomarker or therapeutic target in advanced chronic kidney disease in the future.

## Supporting information

**S1 Data.**
(XLSX)

## Acknowledgments

No further acknowledgments required.

## Author contributions

**Conceptualization:** Rocío Gimena Muñoz, María Pérez Fernández, Rafael Ramírez-Chamond, Diego Rodríguez Puyol, Patricia Martinez Miguel.

**Data curation:** Rocío Gimena Muñoz, Gemma Valera Arévalo, María del Mar Rodríguez San Pedro, María Pérez Fernández, Julia Carracedo, Patricia Martinez Miguel.

**Formal analysis:** Juan Arévalo Serrano, Julia Carracedo.

**Investigation:** Rocío Gimena Muñoz.

**Methodology:** Gemma Valera Arévalo, María del Mar Rodríguez San Pedro, Juan Arévalo Serrano, Diego Rodríguez Puyol, Julia Carracedo.

**Supervision:** Rafael Ramírez-Chamond, Diego Rodríguez Puyol, Patricia Martinez Miguel.

**Validation:** Sushrut S. Waikar, Rafael Ramírez-Chamond, Diego Rodríguez Puyol, Patricia Martinez Miguel.

**Writing – original draft:** Rocío Gimena Muñoz.

**Writing – review & editing:** Sushrut S. Waikar, Diego Rodríguez Puyol, Patricia Martinez Miguel.

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
