## [Editor Report · Decision Letter 0]

11 Mar 2025

Dear Dr. Gimena Muñoz,

Thank you for submitting your manuscript to PLOS ONE. After careful consideration, we feel that it has merit but does not fully meet PLOS ONE’s publication criteria as it currently stands. Therefore, we invite you to submit a revised version of the manuscript that addresses the points raised during the review process.

We look forward to receiving your revised manuscript.

Kind regards,

Daniele Romanello

Academic Editor

PLOS ONE

Journal Requirements:

“NO authors have competing interests”

Additional Editor Comments:

Dear Author,

I decide for a minor revision so you can update a revised version of your paper and abstract with all the needed section.

I hope to start the revision process as soon as you upload the revised version.

Thank you for your work.
---

## [Author Response · Author response to Decision Letter 1]

27 Mar 2025

I have updated the manuscript with the minor revisions your required and included a letter response to reviewers.

---

## [Decision Letter · Decision Letter 1]

22 May 2025

Dear Dr.  Gimena Muñoz,

Thank you for submitting your manuscript to PLOS ONE. After careful consideration, we feel that it has merit but does not fully meet PLOS ONE’s publication criteria as it currently stands. Therefore, we invite you to submit a revised version of the manuscript that addresses the points raised during the review process.

We look forward to receiving your revised manuscript.

Kind regards,

Daniele Romanello

Academic Editor

PLOS ONE

Reviewers' comments:

Reviewer's Responses to Questions

**Comments to the Author**

Reviewer #1: (No Response)

Reviewer #2: (No Response)

2. Is the manuscript technically sound, and do the data support the conclusions?

Reviewer #1: Partly

Reviewer #2: (No Response)

3. Has the statistical analysis been performed appropriately and rigorously?

Reviewer #1: Yes

Reviewer #2: (No Response)

4. Have the authors made all data underlying the findings in their manuscript fully available?

Reviewer #1: Yes

Reviewer #2: (No Response)

5. Is the manuscript presented in an intelligible fashion and written in standard English?

Reviewer #1: Yes

Reviewer #2: (No Response)

Reviewer #1: The true novelty of your study is the possible role of microvesicles, but you haven’t emphasized that in either your Results or Discussion. As evidenced by your review of the existing literature, frailty among patients with CKD has a well-established body of evidence, but that’s what you emphasize in both your Results and Discussion. I recommend that you refocus your report and emphasize your findings regarding microvesicles. Otherwise, your study doesn’t tell us anything that we don’t already know.

Abstract

The results you report here aren’t new except for the association between CMVs and frailty and that’s buried at the end of your results. The median followup isn’t really the point of your study. Report the data from the predictive model (Table 5). That’s far more meaningful than risk factors for frailty that was widely reported.

Introduction

Line 101-102 – Being “highly prevalent” isn’t very meaningful. Report the prevalence and cite your source. For example, you cite a meta-analysis of prevalence by Veronese et al. in your Discussion. What was the prevalence they reported?

Line 108 – The prevalence of frailty in ESRD is important but your focus is CKD. They’re not interchangeable. Can you report the prevalence in CKD as well, if only to provide a baseline for comparison with your own findings? Since your aim is to “analyze frailty” baseline data help readers contextualize your purpose.

Methods

Line 146 – Can you please clarify what you mean by “conservative treatment?” Since it’s an exclusion criteria, readers should have a clear definition.

Lines 150-151 – Please clarify how you measured “3 or more days of asthenia per week?”

Results

Because mortality and dialysis were your end-points, please report the number of patients for each outcome.

Lines 200-202 – Table 1 is not set up to support your statement regarding the prevalence of frailty in patients with type 2 DM and CVD. Table 1 compares frail and non-frail patients, not patients with or without type 2 DM and CVD.

Lines 220-222 –You report using age, T2DM, hemoglobin and total cMVs for the regression model but haven’t reported any differences in cMVs. To justify using total cMVs, it would be helpful to readers for you report differences between frail and non-frail patients before you describe this analysis.

Lines 242-244 – How were the variables for the hazard models chosen? Phosphorus wasn’t significant in the univariate or multivariate models.

Lines 244-245 – You allude to mortality and dialysis here but haven’t reported the actual numbers. Mortality and dialysis data should be reported.

Line 275 – The correlations you report are statistically significant but cannot be described as “strong.” Please see statistical references such as:

Schober P, Boer C, Schwarte LA. Correlation Coefficients: Appropriate Use and Interpretation. Anesth Analg. 2018;126(5):1763-1768.

https://doi.org/10.1213/ANE.0000000000002864

Line 286 – This is a minor point but I believe you have a comma instead of a decimal point for the Nagelkerke square value.

Discussion

You haven’t made good use of your Discussion to highlight the novelty and contribution of your study. Focus on cMVs rather than well known outcomes such as prevalence and risk factors. For example, you discuss phosphorous at length but that wasn’t significant in either the univariate or multivariate model. It doesn’t contribute to the purpose of your study. Tell readers why your findings regarding cMVs are significant. What do they contribute to what we know about frailty?

Line 340 – The correlations you found were statistically significant but they were not “strong.”

Reviewer #2: 1) What exactly is the author's main purpose? To assess the prevalence of frailty in advanced CKD? Or the association between time to dialysis initiation and death?

2) Were the inclusion criteria simply age >18 and eGFR <20? frailty is extremely prevalent in the elderly population, and did the authors consider subgroup comparisons of older and middle-aged patients?

3) Statistical analysis, I would suggest the authors to describe in further detail, especially cox regression, as well as exploring the role of microvesicles

4) Tables 1 and 2 could be combined

5) Table 4 could be considered for replacement with Figure

6) Didn't the article use cox regression? Why do the authors mention in the limitation that “Also, the analysis using estimative binary logistic regression models can be adjusted for a limited number of predictor variables. ”

**Do you want your identity to be public for this peer review?** For information about this choice, including consent withdrawal, please see our Privacy Policy

Reviewer #1: No

Reviewer #2: **Yes: ** Fan Zhang

---

## [Author Response · Author response to Decision Letter 2]

6 Jul 2025

We have attached a Word document with all the responses to the reviewers

---

## [Decision Letter · Decision Letter 2]

2 Sep 2025

Assessment of Frailty in patients with advanced chronic kidney disease, and the role of Microvesicles: A Single-Center Study

PONE-D-24-44013R2

Dear Dr. Gimena Muñoz,

We’re pleased to inform you that your manuscript has been judged scientifically suitable for publication and will be formally accepted for publication once it meets all outstanding technical requirements.

Kind regards,

Miquel Vall-llosera Camps

Senior Staff Editor

PLOS One

Reviewers' comments:

Reviewer's Responses to Questions

**Comments to the Author**

Reviewer #1: All comments have been addressed

Reviewer #2: All comments have been addressed

2. Is the manuscript technically sound, and do the data support the conclusions?

Reviewer #1: Yes

Reviewer #2: (No Response)

3. Has the statistical analysis been performed appropriately and rigorously?

Reviewer #1: Yes

Reviewer #2: (No Response)

4. Have the authors made all data underlying the findings in their manuscript fully available?

Reviewer #1: Yes

Reviewer #2: (No Response)

5. Is the manuscript presented in an intelligible fashion and written in standard English?

Reviewer #1: Yes

Reviewer #2: (No Response)

Reviewer #1: Thank you for your detailed responses to my questions and comments. Your revisions are both thoughtful and effective, and as a result your manuscript now highlights the true novelty of your findings. In particular, the 41% variability in frailty explained by the three variables is impressive. There are just two points in the revised wording that I believe merit your further consideration.

Methods (line 156) – The structured questionnaire used for assessment of asthenia is intriguing. Weakness in frailty is typically assessed through objective measures such as hand grip strength. A self-report questionnaire would likely be of interest to readers. If the questionnaire you used has been previously reported or perhaps validated, you might cite those studies for interested readers.

Discussion (line 320-322) – The wording of this sentence is somewhat confusing. I think it’s just an English translation issue. Table 1 compares age by frailty status (frail vs. non-frail). It doesn’t compare frailty by age. The comparator (the “by XX” characteristic) is represented by the grouping along the X-axis, which in this case is frailty status.

Reviewer #2: The author has well explained and corrected the doubts raised by the reviewers. However, it is hoped that in future work, the sample size can be expanded and more stable statistical analysis methods can be adopted

**Do you want your identity to be public for this peer review?** For information about this choice, including consent withdrawal, please see our Privacy Policy

Reviewer #1: **Yes: ** Melissa J. Benton, PhD, RN, FACSM, FGSA

Reviewer #2: No

---

## [Editor Report · Acceptance letter]

PONE-D-24-44013R2

PLOS ONE

Dear Dr. Gimena Muñoz,

I'm pleased to inform you that your manuscript has been deemed suitable for publication in PLOS ONE. Congratulations! Your manuscript is now being handed over to our production team.

Kind regards,

on behalf of

Dr. Miquel Vall-llosera Camps

Staff Editor

PLOS ONE